# A Non-Canonical Teleost NK-Lysin: Antimicrobial Activity via Multiple Mechanisms

**DOI:** 10.3390/ijms232112722

**Published:** 2022-10-22

**Authors:** Hang Xu, Zihao Yuan, Li Sun

**Affiliations:** 1CAS and Shandong Province Key Laboratory of Experimental Marine Biology, Institute of Oceanology, Center for Ocean Mega-Science, Chinese Academy of Sciences, Qingdao 266071, China; 2Laboratory for Marine Biology and Biotechnology, Pilot National Laboratory for Marine Science and Technology, Qingdao 266237, China; 3College of Earth and Planetary Sciences, University of Chinese Academy of Sciences, Beijing 100049, China

**Keywords:** NK-lysin, *Paralichthys olivaceus*, antimicrobial peptide, bactericidal

## Abstract

NK-lysin (NKL) is a family of antimicrobial proteins with an important role in innate and adaptive immunity. In this study, a non-canonical NK-lysin (NKLnc) was identified in the Japanese flounder (*Paralichthys olivaceus*), which shares low sequence identities (15.8–20.6%) with previously reported fish NKLs and was phylogenetically separated from the canonical NKLs in teleost. NKLnc expression was upregulated in flounder tissues during bacterial infection, and interference with NKLnc expression impaired the ability of flounder cells to eliminate invading bacteria. When expressed in *Escherichia coli*, NKLnc was detrimental to the host cells. P35, a peptide derived from the saposin B domain (SapB) of NKLnc, bound major bacterial surface molecules and killed both Gram-negative and Gram-positive bacteria by inflicting damage to bacterial cell structure and genomic DNA. The bactericidal activity, but not the bacteria-binding capacity, of P35 required the structural integrity of the alpha 2/3 helices in SapB. Furthermore, P35 induced the migration of flounder peripheral blood leukocytes, inhibited bacterial dissemination in fish tissues, and facilitated fish survival after bacterial challenge. Together our study reveals that NKLnc plays an important part in flounder immune defense, and that NKLnc peptide exerts an antimicrobial effect via multiple mechanisms by targeting both bacteria and fish cells.

## 1. Introduction

The emergence of multidrug-resistant bacteria and the prevalence of large-scale antimicrobial resistance, mainly caused by the misuse of traditional antibiotics, have become a serious threat to public health and environmental safety [1,2,3]. As a result, it is imperative to search for new antimicrobial agents that are less prone to resistance development. One of the promising alternatives to conventional antibiotics is gene-coded natural antimicrobials represented by antimicrobial peptides (AMPs). AMPs are evolutionarily conserved innate immune factors present in almost all life domains [4]. Generally, AMPs are short (<100 amino acids), amphiphilic peptides with abundant positively charged amino acid residues and hydrophobic residues [5]. AMPs constitute the first-line of immune defense against the infection of a broad spectrum of bacteria, fungi, parasites, and viruses [5]. The antimicrobial mechanisms of AMPs depend largely on their ability to directly alter or destroy the cytoplasmic membranes of the target organisms [6]. In addition, AMPs can also target intracellular molecules such as DNA and protein [6,7,8,9]. To date, nearly 1900 AMPs have been identified or chemically synthesized based on the sequences of diverse living organisms, including microbes, insects, plants, fish, amphibians, birds, and mammals [5,10,11,12,13,14]. Owing to their biochemical and pharmacodynamic properties, AMPs are much less prone to resistance development compared to conventional antibiotics and, therefore, are good candidates for translational applications [15].

NK-lysin (NKL), or granulysin in humans, is a type of potent AMP that belongs to the saposin (Sap)-like protein superfamily [14,16]. In human, granulysin is secreted, together with the pore forming protein perforin, from the granules of cytotoxic T lymphocytes (CTLs). Granulysin is synthesized as a precursor protein that is processed by an unknown protease to generate a 9 kDa protein with pore forming activity [17]. Similar to most AMPs, the 9 kDa granulysin is highly cationic, but the molecular mechanism of its pore formation is unclear [17]. Structurally, NKL homologues possess a SapB domain and amphipathic α-helical structures. The human granulysin adopts five alpha helices, with α2 and 3 being functionally essential [17]. In addition to human granulysin, NKL homologues from pig, chicken, and cattle were studied, which showed that the peptides derived from NKLs exhibited antimicrobial activities against various microbes [14,18,19]. Some of the NKL peptides interacted with the bacterial plasma membrane and formed transmembrane pores that led to bacterial death [16,20].

In fish, studies of NKL homologues were reported in a number of teleost, including zebrafish (*Danio rerio*), tongue sole (*Cynoglossus semilaevis*), common carp (*Cyprinus carpio*), turbot (*Scophthalmus maximus*), Atlantic salmon (*Salmo salar*), Nile tilapia (*Oreochromis niloticus*), rainbow trout (*Oncorhynchus mykiss*), large yellow croaker (*Larimichthys crocea*), channel catfish (*Ictalurus punctatus*), yellow catfish (*Pelteobagrus fulvidraco*), golden pompano (*Trachinotus ovatus*), European sea bass (*Dicentrarchus labrax*), and mudskipper (*Boleophthalmus pectinirostris*) [21,22,23,24,25,26,27,28,29,30,31,32,33]. Similar to mammalian NKLs, fish NKLs possess the conserved SapB domain with the ability to form α-helix. Based on the NKL sequences, a number of antimicrobial peptides were synthesized and shown to display antimicrobial activity.

In Japanese flounder (*Paralichthys olivaceus*), a NKL homologue was reported in 2007, and a NKL-based peptide (JF-NK-2) was shown to inhibit the growth of Gram-negative bacteria (*Escherichia coli*, *Klebsiella pneumoniae*, *Pseudomonas aeruginosa*, and *Photobacterium damselae subsp. piscicida*), but had no effect on the growth of Gram-positive bacteria [34]. In the present study, we identified a non-canonical NKL (named NKLnc) from the Japanese flounder that shares low sequence identities with the previously reported fish NKLs, including the flounder NKL described above. We examined the structure, expression, and antimicrobial properties of NKLnc and investigated the bactericidal activity and mechanism of NKLnc-derived peptides in vitro and in vivo.

## 2. Results

### 2.1. Identification of a Non-Canonical NKL in Flounder

A NK-lysin ortholog was identified in the genome of the flounder. The encoded protein sequence shared 18.1% identity with the flounder NKL reported in 2007 [34]. Phylogenetic analysis showed that the newly identified NKL was excluded from the clade formed by the canonical fish NKLs, which include almost all the reported fish NKLs (Figure 1A). Based on this observation, we named this flounder NKL non-canonical NKL (NKLnc). NKLnc contains 134 amino acids with a calculated molecular weight of 15.06 kDa and an isoelectric point (pI) of 9.68. It contains an N-terminal signal peptide (1–19 aa) and a SapB domain (57–132 aa), in which six cysteine residues were predicted to form three disulfide bonds (C58-C132, C61-C126, and C89-C94) (Figure 1B). Multiple sequence alignment showed that NKLnc shares 15.8–20.1% sequence identities with the canonical NKLs. NKLnc also differs from the canonical NKLs at the signal peptide cleavage site (Figure 1B).

### 2.2. NKLnc Expression Is Regulated by Bacterial Pathogen and Required for Effective Blocking of Bacterial Infection

RT-qPCR showed that NKLnc expression was detected in eight issues of flounder, with the highest levels found in the spleen and kidney and the lowest levels in the muscle and brain (Figure 2A). NKLnc expressions in the gill, blood, and liver were abundant, though lower than that in the spleen and kidney. Upon infection by *Vibrio harveyi* (a common fish pathogen), NKLnc expression was significantly altered in the kidney, spleen, and liver of the flounder (Figure 2B). In the kidney and spleen, NKLnc expression was significantly increased at 6, 12, 24, and 48 hpi, with the peak expression occurring at 12 hpi. In the liver, NKLnc expression peaked at 6 hpi and then rapidly declined to a normal level. To determine the role of NKLnc in antimicrobial immune defense, NKLnc expression in flounder cells (FG-9037 cells) was knocked down by RNAi before *V. harveyi* infection. Subsequent bacterial recovery analysis showed that the bacterial loads in FG-9037 cells with NKLnc knockdown were significantly higher than that in the control cells (Figure 2C).

### 2.3. NKLnc Expression Is Detrimental to Host Bacteria

To examine the potential antimicrobial effect of NKLnc, the NKLnc gene was introduced into *E. coli* in a manner that NKLnc expression was inducible by IPTG (isopropyl-β-D-thiogalactoside). In the presence of IPTG, the growth of the NKLnc-expressing *E. coli* was dramatically inhibited compared to that of the control *E. coli* (Figure 3A). Consistently, the number of NKLnc-expressing *E. coli* cells was much lower than that of the control cells (Figure 3B,C). These results indicated that expression of NKLnc was lethal or detrimental to the host bacterial cells.

### 2.4. A NKLnc-Derived Peptide Is Bactericidal in a Manner That Depends on Two Alpha Helices of SapB

Since, in human granulysin, the α2 and α3 helices in the core region of the SapB domain are functionally essential, two peptides (P35 and P17) corresponding to these alpha helices in the SapB of NKLnc were synthesized. P35 is composed of 35 residues and derived from the α2 and α3 of the SapB domain, while P17 is a truncated version of P35 and lacks α3 (Figure 4A). P35 and P17 were predicted to adopt two and one α helix, respectively. In the helical wheel views of P35 and P17, the predominant polar residues and hydrophobic residues were clustered separately (Figure 4B). Electrostatic surface potential showed that P35 exhibited more abundant positive charge than P17 (Figure 4C), indicating a stronger cationic nature of P35. P35 inhibited the growth of both Gram-negative and Gram-positive bacteria, including the common fish pathogens *Vibrio anguillarum*, *Vibrio harveyi*, and *Streptococcus iniae* (Table 1). The most potent inhibitory activity was detected against *V. anguillarum* and *B. subtilis*. P35 killed *V. anguillarum*, *V. harveyi,* and *S. iniae* but had no apparent effect on the viability of *E. tarda* (Figure 4D). P35 bound to Gram-positive and Gram-negative bacteria, the latter including *E. tarda*, in a dose-dependent manner (Figure 4E). Consistently, P35 bound to bacterial cell wall components of both Gram-negative and Gram-positive bacteria, i.e., peptidoglycan (PGN) and lipoteichoic acid (LTA), and lipopolysaccharide (LPS) (Figure 4F). Compared to P35, P17 exhibited much weaker antimicrobial activities (Table 1, Figure 4D) but similar binding activities towards bacterial cell wall components and bacterial cells (Appendix A).

### 2.5. P35 kills Bacteria by Damaging Bacterial Plasma Membrane and Chromosomal DNA

To investigate the bactericidal mechanism of P35, we first examined the effect of P35 on the cellular integrity of the target bacteria. Propidium iodide (PI) staining showed that *V. harveyi* treated with P35 became highly susceptible to PI (Figure 5A), implying cytoplasm membrane disruption. Consistently, scanning electron microscopy (SEM) and transmission electron microscopy (TEM) revealed that P35-treated *V. harveyi* exhibited broken cellular structure and much lighter intracellular density due to loss of cellular contents (Figure 5B,C). We next examined the effect of P35 on the stability of bacterial genomic DNA (gDNA). When treated with P35 in vitro, the gDNA of *V. harveyi* degraded in a manner that correlated with the dose of P35 (Figure 5D). Likewise, when live *V. harveyi* was treated with P35, the intracellular gDNA was degraded in a time dependent manner (Figure 5E).

### 2.6. P35 Possesses Chemotactic Activity and Promotes Bacterial Clearance from Fish Tissues

In addition to bacteria, we also examined the effect of P35 on fish immune cells and fish immune defense. We found that the presence of P35 induced the migration of flounder PBL in a dose-dependent manner (Figure 6A). When flounder were infected with *V. harveyi* in the presence of P35, the bacterial loads in the kidney, spleen, and liver significantly decreased at 36 hpi (Figure 6B). Consistently, the presence of P35 delayed the onset of mortality in the flounder and prolonged the survival of the fish. Although, similar to the control fish, P35-treated fish exhibited 100% mortality after prolonged infection, the death process of P35-treated fish was significantly slowed compared to that of the control fish (Figure 6C).

## 3. Discussion

To date, NK-lysin homologues were reported in at least 14 fish species, which, with the exception of the NKL from large yellow croaker (LCNKL2), are closely related and belong to the same phylogenetic group. These canonical fish NKLs include the flounder NKL reported in 2007 [34]. In the present study, we identified and characterized a non-canonical NKL, NKLnc. NKLnc possesses structural features conserved in human granulysin and mammalian/teleost NKLs, including the SapB domain and the six disulfide bond-forming cysteine residues, which define NKLnc as a member of the NK-lysin family. However, NKLnc shares low sequence identities with the canonical fish NKLs and belongs to a phylogenetic group distinctly separated from that formed by the canonical fish NKLs. In addition, the signal peptide cleavage site of NKLnc is markedly different from that of canonical NKLs. These characteristics indicate that NKLnc represents a novel type of fish NKL. Nevertheless, the presence in NKLnc of the conserved structures suggests the conservation of basic NKL functions.

In zebrafish, there exist four NKL genes (NKLa, b, c, and d) in the genome. These orthologues exhibited different patterns of tissue specific expression and differed in response to the infection of Spring Viraemia of Carp Virus, which promoted the expression of NKLa and d but not the other two orthologues [21]. In large yellow croaker, two NKL homologs, LcNKL1 and LcNKL2, were identified that belong to canonical and non-canonical NKL, respectively. LcNKL1 and LcNKL2 displayed similar expression profiles in several tissues but not in the brain, in which LcNKL2 was undetectable [32,35]. During *Cryptocaryon irritan*s infection, LcNKL1 and LcNKL2 showed similar expression patterns in immune related tissues [32,35]. In flounder, the expression of the canonical NKL was relatively high in the gills, heart, head kidney, intestines, and spleen and low or absent in the liver and was not inducible by LPS [34]. The constitutive expression pattern of NKLnc was largely similar to that of the canonical flounder NKL in most tissues, except the liver, in which NKLnc expression was abundant. During *V. harveyi* infection, NKLnc expression was significantly upregulated in flounder tissues, indicating involvement of NKLnc in pathogen-induced immune response. In agreement, interference with the normal NKLnc expression in flounder cells significantly reduced the ability of the cells to eliminate invading *V. harveyi*. These results support an antibacterial role of NKLnc in flounder.

NKL peptides are known to kill various microorganisms [22,25,28,32,36]. JF-NKL-2, the peptide based on the canonical flounder NKL, inhibited the growth of Gram-negative bacteria but had no effect on Gram-positive bacteria [34]. In the present study, we found that P35 derived from NKLnc bound well to typical pathogen associated molecular pattern (PAMP) of both Gram-positive and Gram-negative bacteria. Consistently, P35 bound and killed bacteria of both Gram-positive and Gram-negative natures. Plasma membrane disruption is a common antimicrobial mechanism utilized by many AMPs. Likewise, for P35, both PI staining and electron microscopy revealed that P35 increased the membrane permeability of the target bacteria, which led to loss of membrane potential, collapse of bacterial cellular structure, and release of the intracellular contents. In addition to targeting the bacterial membrane, P35 was also found to induce degradation of bacterial DNA both in vitro and in vivo. It is likely that the highly cationic nature of the peptide likely contributed to the interaction with DNA.

Alpha helix is broadly present in many AMPs, and some AMPs are composed of a single helix [37]. Human granulysin has five α helices, of which, α2 and α3 are vital to their function. In this study, we found that NKLnc could adopt similar structures, and that P35, which contains the region corresponding to the α2 and 3 helices and was predicted to form two α helices, displayed effective bactericidal activity, whereas P17, which retains the α2 region but lacks the α3 region, exhibited markedly reduced bactericidal activity, suggesting an essential role of the α3 region in P35 activity. This is probably due to the fact that α3 not only confers on P35 the higher (compared to P17) cationic electrostatic surface potential but also provides a conserved Cys residue that enables the formation of a disulfide linkage within the peptide. However, despite its impaired bactericidal activity, P17 bound well to bacteria and PAMPs, implying that the α3 region is not required for bacterial/PAMP interaction. Since P17 still retains an amphipathic structure conferred by the α2 region, it is likely that P17 could exert electrostatic interaction with the anionic bacterial surface molecules, which led to stable bacterial binding but not further penetration into and disruption of the plasma membrane.

In addition to acting as bactericides, AMPs can also play other roles. For example, human cathelicidin LL-37 is able to regulate cell migration and modulate immune response and biofilm development [38,39]. In our study, we found that P35 could direct the trafficking of flounder PBL, implying a chemotactic property. This property, together with the bactericidal activity, endowed P35 with the potential of an effective in vivo antimicrobial. Indeed, in vivo infection showed that when flounder were infected with *V. harveyi* in the presence of P35, the bacterial burdens in fish tissues were significantly reduced, and fish survival was prolonged.

## 4. Materials and Methods

### 4.1. Animals 

Clinically healthy Japanese flounder (*Paralichthys olivaceus*) were purchased from a commercial farm. The fish were acclimatized in the laboratory, as reported previously [40], for one week prior to the study. When tissue dissection was involved, the fish were anesthetized with tricaine methane sulfonate (Sigma, St. Louis, MO, USA). 

### 4.2. Bacteria and Cell Line

The bacteria used in this study were reported previously [41]. *Vibrio anguillarum*, *Vibrio harveyi*, *Pseudomona fluorescens,* and *Edwardsiella tarda* were cultured in Luria-Bertani (LB) broth at 28 °C. *Streptococcus iniae* was cultured in Tryptic Soy Broth (TSB) at 28 °C. *Bacillus cereus* and *Bacillus subtilis* were cultured in marine 2216E medium at 28 °C. *Escherichia coli*, *Micrococcus luteus,* and *Staphylococcus aureus* were cultured in LB broth at 37 °C. All bacteria were grown overnight in appropriate media and temperature; the cultures were then diluted 1:100 in fresh medium and cultured to the mid-logarithmic phase. The bacteria were harvested by centrifugation and washed three times with PBS. The Japanese flounder cell line FG–9307 [42] was cultured at 24 °C in L-15 medium (Sigma, St Louis, MO, USA) containing 10% FBS (ExCell Bio, Shanghai, China), 100 units/mL penicillin, and 100 µg/mL streptomycin.

### 4.3. Sequence and Structure Analyses

All gene and protein sequences were obtained from the National Center for Biotechnology Information (NCBI). The GenBank accession number of NKLnc is XP_019940439.1. The signal peptide cleavage site was predicted with SignalP 4.1 (http://www.cbs.dtu.dk/services/SignalP/). The disulfide bond was predicted with Prosite (http://prosite.expasy.org/). The protein domain and secondary structure were predicted using SMART (http://smart.emblheidelberg.de/) and Jpred 4 (http://www.compbio.dundee.ac.uk/jpred/index.html). The multiple sequence alignment was performed using ClustalW (http://clustalw.ddbj.nig.ac.jp/) and generated using ESPript 3.0 (http://espript.ibcp.fr/ESPript/cgi-bin/ESPript.cgi). The phylogenetic analysis was conducted using MEGA7. The three-dimensional (3D) models of NKLs were predicted using Iterative Threading Assembly Refinement (https://zhanglab.ccmb.med.umich.edu/I-TASSER/) and visualized with PyMOL (PyMOL Molecular Graphics System, Version 2.5.0, Schrödinger, Tokyo, Japan) [43]. Electrostatic surface potentials were determined with Adaptive Poisson–Boltzmann Solver (APBS) and visual molecular dynamics (VMD) [44]. Peptide helical wheels were generated with HeliQuest (https://heliquest.ipmc.cnrs.fr/cgibin/ComputParams.py). The NKL sequences used for phylogenetic analysis and sequence alignment are listed in Appendix A.

### 4.4. Quantitative Real Time PCR (qRT-PCR)

To determine NKLnc expression in healthy fish tissues, the brain, gill, intestine, kidney, spleen, liver, blood, and muscle were dissected from healthy flounder. Total RNA extraction, cDNA synthesis, and qRT-PCR were carried out as previously reported [45]. To determine NKLnc expression during bacterial infection, flounder were injected intraperitoneally with 100 µL 1 × 10^6^ CFU *V. harveyi* in PBS or 100 µL PBS (control). At 6, 12, 24, and 48 h post-infection (hpi), the kidney, spleen and liver were collected. β-actin was used as an internal reference for healthy fish, and 18S rRNA, α-tubulin, and GAPDH were used as internal references for the fish with bacterial infections [46]. The primers used are listed in Appendix A.

### 4.5. The Effect of NKLnc Expression on the Growth and Survival of Host Bacteria

To determine the effect of NKLnc expression on host bacterial cells, the coding sequence of NKLnc was PCR amplified with primers NKLnc-F/R (Appendix A), and the PCR product was inserted into pET-30a (+) vector (Novagen, Madison, WI, USA). The resulting recombinant plasmid was introduced into *E. coli* Transetta (DE3) (TransGen, Beijing, China). The transformant was cultured to OD600 0.6–0.8 in LB medium and the culture was then divided into groups. Isopropyl β-D-1-thiogalactopyranoside (IPTG) (0.3 mM) was added into one group to induce NKLnc expression. The two groups were cultured for various hours, and bacterial growth at different time points was recorded by measuring OD600. For plate counting, the above *E. coli* transformant was plated on LB agar plates supplemented with or without 0.3 mM IPTG. After incubation at 37 °C overnight, the colony numbers were calculated.

### 4.6. Peptides

First, 5′-FITC labeled P17 (5′-KLIRQACNKIIGHFKRK-3′) and P35 (5′-SKEKIDRLLNKACNGIKCKLIRQACNKIIGHFKRK-3′) were chemically synthesized by Science Peptide Biological Technology Co., Ltd. (Shanghai, China). The negative control peptide (NCP) P86P15 [23,36] was synthesized similarly. The peptides were purified by high-performance liquid chromatography to 95% of purity. Before use, the peptides were dissolved in PBS.

### 4.7. Effect of NKLnc Knockdown on Bacterial Infection in Flounder Cells

The siRNAs used in this study were synthesized by GenePharma (Shanghai, China). The sequences of the siRNAs are listed in Appendix A. Gene knockdown by RNA interference was performed as described previously [47]. Briefly, FG-9307 cells were transfected with or without (control) the NKLnc-specific siR1 or siR2, or the negative control siRNA (NCS) for 24 h using Lipofectamine RNAiMAX (Invitrogen, Carlsbad, CA, USA). Gene knockdown was verified by qRT–PCR (Appendix A). The NKLnc-knockdown cells and control cells were infected with *V. harveyi,* as reported previously [47] with slight modification. Briefly, *V. harveyi* was prepared as described above and resuspended in PBS. The bacteria were added to the above FG-9307 cells in a 96-well plate (Nest Biotechnology, Wuxi, China) at a multiplicity of infection (MOI) of 3:1. At 2, 4, and 8 hpi, the cells were lysed with 1% Triton X–100, and the lysate was plated onto LB agar plates. The plates were incubated at 28 °C for 24 h, and the number of colonies was counted.

### 4.8. Antibacterial Activity of NKLnc Peptides

The binding of the peptides to bacteria and bacterial cell wall components, i.e., lipopolysaccharide (LPS), peptidoglycan (PGN) and lipoteichoic acid (LTA), was determined by enzyme-linked immunosorbent assay (ELISA), as described previously [48]. The minimal inhibitory concentration (MIC) and bactericidal activity of the peptides were determined as reported previously [49]. Briefly, the bacteria were diluted in medium to a final concentration of 1.0 × 10^5^ CFU/mL, and a 100 μL aliquot of the bacteria suspension was added to 96-well plates. Serial dilutions of the peptides (80, 40, 20, 10, 5, 2.5 µM) were prepared, and an aliquot of 100 μL of each dilution was added to above 96-well plates. The plates were incubated at appropriate temperatures for 24 h. MIC was determined by visual inspection of bacterial growth.

### 4.9. Propidium Iodide (PI) Uptake

*V. harveyi* was cultured as above and resuspended in PBS to 1 × 10^7^ CFU/mL. Peptide or PBS (control) was mixed with 100 µL *V. harveyi* suspension in a 96-well plate. After incubation at room temperature for 2 h, the cells were stained with PI (Invitrogen, Carlsbad, CA, USA) for 5 min in the dark. The bacteria were observed with a microscope.

### 4.10. Electron Microscopy

*V. harveyi* was prepared as described above and resuspended in PBS to 1 × 10^8^ CFU/mL. P35 or NCP was added to the bacterial suspension at a final concentration of 10 µM. After incubation at room temperature for 2 h, the bacteria were fixed with 2.5% glutaraldehyde in PBS, followed by washing three times with PBS. The bacteria were dehydrated through an increasing concentration of ethanol (30, 50, 70, 80, 90 and 100%). The bacteria were resuspended in isopentyl acetate and observed with a scanning electron microscope (SEM) (Hitachi, S-3400N, Tokyo, Japan) and a transmission electron microscope (TEM) (Hitachi, HT7700, Japan).

### 4.11. Effect of NKLnc on Bacterial Genomic DNA (gDNA)

Bacterial gDNA was extracted using the Bacterial DNA kit (Sparkjade Biotechnology Co. Ltd., Shandong, China) according to the instructions of the manufacturer. Degradation of *V. harveyi* gDNA by P35 was assayed as previously described [36]. Briefly, for in vitro assay, 1 µg gDNA was mixed with an increasing amount (0, 1.25, 2.5, 5, 10, 20, and 40 µM) of P35 or NCP, followed by incubation at room temperature for 1 h. The mixture was applied to a 1.2% agarose gel electrophoresis. For the in vivo assay, *V. harveyi* was suspended in PBS to 1.0 ×10^8^ CFU/mL. P35 or NCP was added to the bacterial suspension (100 µL) at a final concentration of 10 µM and incubated at 28 °C for 0, 1, 2, 4, or 8 h. After incubation, *V. harveyi* gDNA was extracted and examined as above by electrophoresis.

### 4.12. Chemotaxis Assay

Japanese flounder peripheral blood leukocytes (PBL) were prepared as reported previously [39]. Briefly, the blood of the flounder was placed on top of 61% Percoll (Solarbio, Beijing, China) and centrifuged at 600 g for 10 min. The layer of PBL was collected and washed with PBS. The cells were then resuspended in L-15 medium. The chemotaxis assay was performed as reported previously [50]. P35 or NCP was diluted to 0.25~10 µM with L15 medium, and 500 μL of the diluted peptide was added into the lower wells of a 24-well Transwell plate (Corning Costar Co., Cambridge, MA, USA), which were covered with nitrocellulose filter membrane. Then, PBL were added to the upper chamber and incubated at room temperature for 4 h. The cells that migrated to the lower chamber were examined using a microscope.

### 4.13. In Vivo Effect of P35 on Bacterial Infection and Host Survival

To examine the effect of P35 on bacterial dissemination in fish tissues, P35 and NCP were resuspended in PBS to 20 µg/mL. *V. harveyi* was cultured as above and resuspended in PBS, P35, or NCP to 1.0 × 10^7^ CFU/mL. Three groups of flounder (n = 12) were injected intraperitoneally with 100 µL *V. harveyi*, *V. harveyi*-P35 mixture, or *V. harveyi*-NCP mixture. At 12 and 36 hpi, the kidney, liver and spleen were collected from the fish (5 fish/time point) and examined for bacterial recovery by plate count. The effect of P35 on fish survival after infection was performed as previously described [51]. The flounder were randomly divided into three groups (n = 30). The fish were infected as above with *V. harveyi* (1.0 × 10^6^ CFU/fish) in the absence (control) or presence of P35 or NCP or P35. The fish were monitored daily for mortality.

### 4.14. Statistical Analysis

Statistical analyses were performed with GraphPad Prism 7 (www.graphpad.com/). Student’s t test and one-way analysis of variance (ANOVA) were used for comparisons between groups. Log-rank was used for the analysis of fish survival. Statistical significance was defined as *p* < 0.05.

## 5. Conclusions

In this study, we identified a novel fish NKL and demonstrated its participation in pathogen-induced immune response and optimal defense against bacterial infection in flounder. The NKLnc peptide is both a structure-dependent AMP and a chemotactic factor that kills bacteria in vitro and reduces bacterial infection in vivo. The multi-action mechanism of P35 indicates a low possibility of resistance development in bacteria. These results, along with the observation that P35 has a strong bactericidal effect against the most common and severe aquaculture pathogens of a wide variety of fish and invertebrate marine animals, suggest an application potential of P35 in the control of bacterial diseases in aquaculture and other relevant fields.

## Figures and Tables

**Figure 1 ijms-23-12722-f001:**
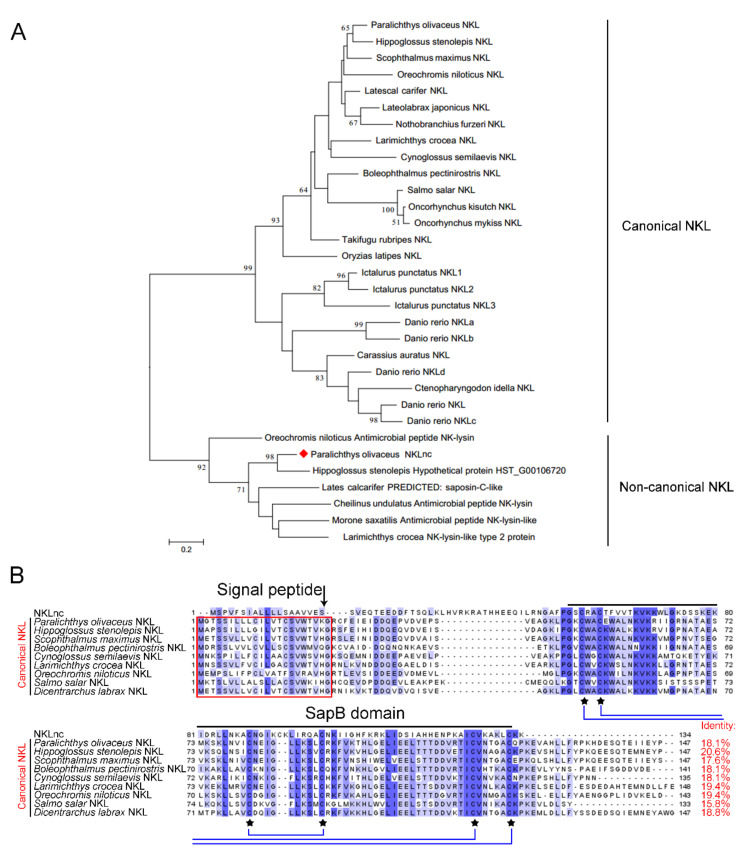
Phylogenetic and sequence analyses of NKLnc. (**A**) Phylogenetic analysis of NKLnc. The values at the forks indicate the percentage of trees in which this grouping occurred after bootstrapping (1000 replicates; shown only when the values were >50%). Scale bar, the number of substitutions per base. (**B**) Sequence alignment of fish NKL homologues. The residues that are >50% identical are shaded, with identical residues shaded in dark blue. The alignment gaps are indicated with “-”. The predicted cleavage site for the signal peptide is indicated with “↓”. The conserved six cysteine residues are marked with “★”. The cysteine residues that form a disulfide bond are linked by a blue line.

**Figure 2 ijms-23-12722-f002:**
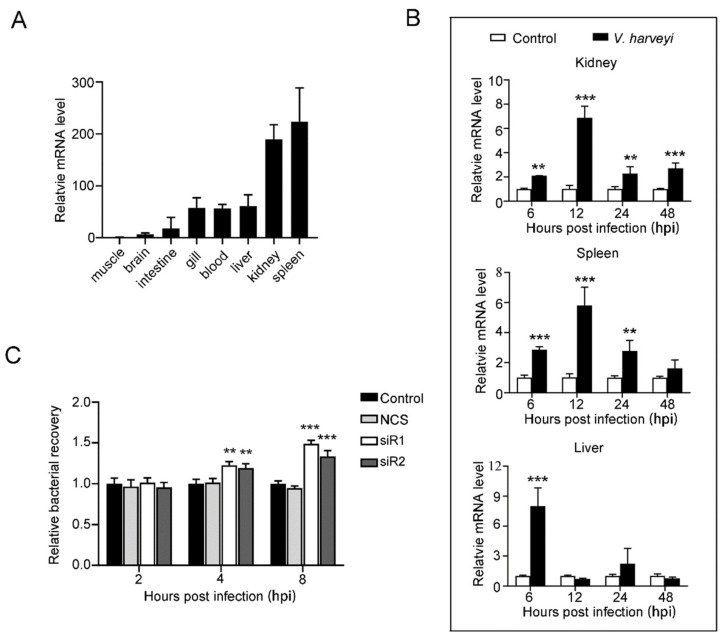
NKLnc expression and its effect on bacterial infection. (**A**) NKLnc expression in flounder tissues under normal physiological conditions was determined by RT-qPCR. The expression levels are presented relative to that in the muscle. (**B**) Flounder were infected with or without (control) *Vibrio harveyi* for different numbers of hours, and the expression of NKLnc in the kidney, spleen, and liver was determined by RT-qPCR. (**C**) FG-9307 cells were treated with or without (control) NKLnc-targeting siRNA (siR1 or siR2) or the negative control siRNA (NCS) and then infected with *V. harveyi* for different numbers of hours. The bacterial recovery was determined. For panels (**B**,**C**), values are the means of replicate experiments and shown as means ± S.D. *** *p* < 0.001; ** *p* < 0.01.

**Figure 3 ijms-23-12722-f003:**
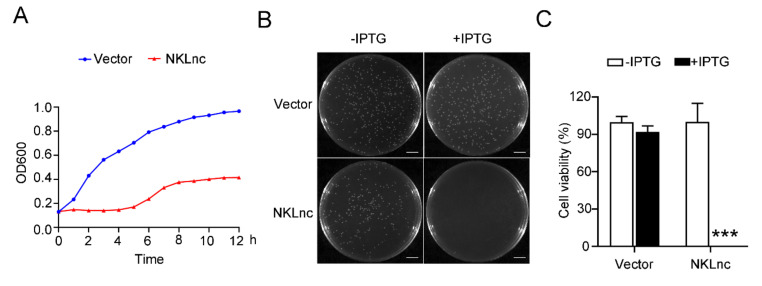
The effect of NKLnc expression on host bacterial growth. (**A**) *Escherichia coli* was transformed with the backbone vector or the vector carrying the NKLnc gene inducible by IPTG. The bacteria were cultured in the presence of IPTG and determined for growth at different hours. (**B**,**C**) The above *E. coli* cells were plated in an agar plate supplemented with or without IPTG, and bacterial growth was observed after overnight incubation. Scale bar, 1 cm (**B**). The percentages of bacterial survival were determined (**C**). Values are the means of triplicate assays and shown as means ± S.D. *** *p* < 0.001.

**Figure 4 ijms-23-12722-f004:**
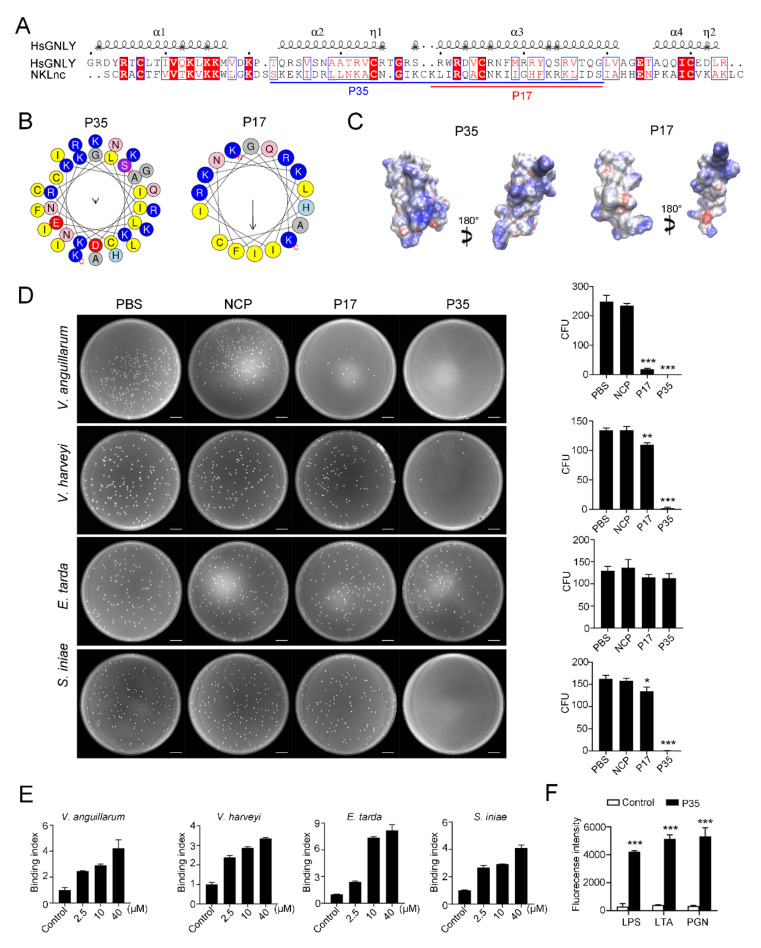
NKLnc peptide structure and antimicrobial activity. (**A**) Secondary-structure-based alignment of the SapB domain sequences in human granulysin (HsGNLY) and NKLnc. The conserved amino acid residues are highlighted in red (100% identical) or boxed. The secondary structure is indicated on the top of the aligned sequence. P35 and P17 are underlined in blue and red, respectively. (**B**,**C**) The helical wheels (**B**) and electrostatic surface potentials (**C**) of P35 and P17. In (**B**), the residues that are positively charged, negatively charged, and hydrophobic are in blue, red, and yellow, respectively. In (**C**), the positive- and negative-charged regions are in blue and red, respectively. (**D**) *Vibrio anguillarum*, *Vibrio harveyi*, *Edwardsiella tarda,* and *Streptococcus iniae* were incubated with P17, P35, the negative control peptide (NCP), or PBS for 2 h. Bacterial survival was determined by plate count. The bacterial numbers (shown as colony forming unit, CFU) are shown on the right panels. Scale bar, 1 cm. (**E**) Bacteria binding to different doses of P35 or control peptide (40 μM) was determined by ELISA. (**F**) LPS, LTA, and PGN binding to P35 or control peptide were determined by ELISA. For panels (**D**–**F**), values are the means of triplicate experiments and shown as means ± S.D. *** *p* < 0.001; ** *p* < 0.01; * *p* < 0.05.

**Figure 5 ijms-23-12722-f005:**
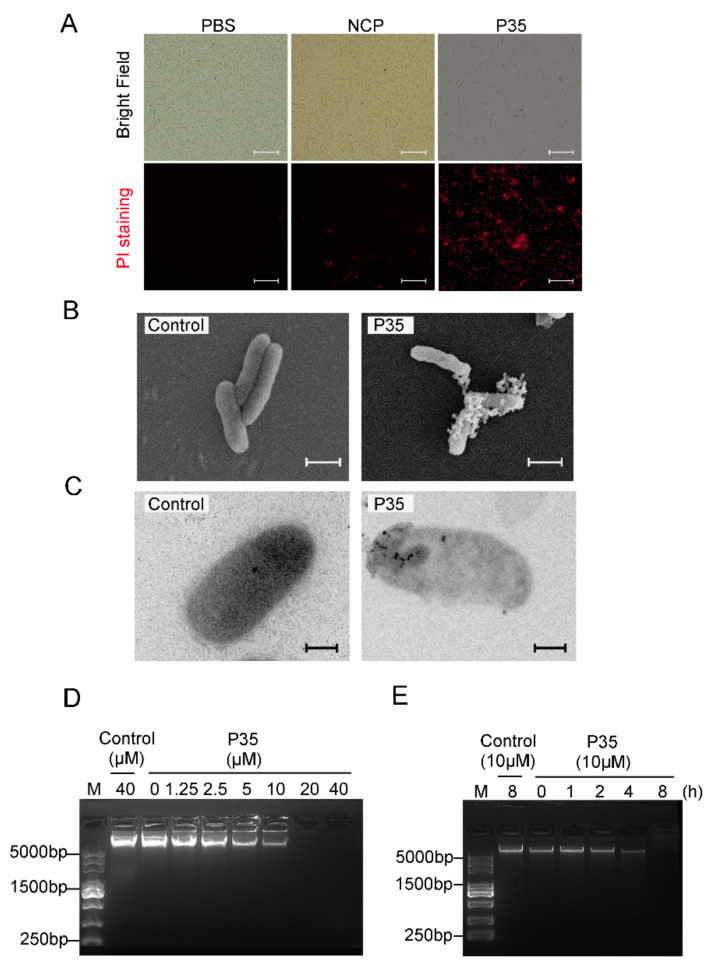
The damaging effect of P35 on bacterial structure and DNA. (**A**–**C**) *Vibrio harveyi* was incubated with P35, the negative control peptide (NCP), or PBS for 2 h and then subjected to PI staining (**A**), scanning electron microscopy (**B**), and transmission electron microscopy (**C**). Scale bars, 30 μm (**A**), 1 μm (**B**), and 200 nm (**C**). (**D**) *V. harveyi* genomic DNA was incubated with different concentrations of P35 or with control peptide for 1 h and then subjected to electrophoresis. (**E**) *V. harveyi* was incubated with P35 for 0 to 8 h or with control peptide for 8 h. The genomic DNA of the bacteria was subjected to electrophoresis.

**Figure 6 ijms-23-12722-f006:**
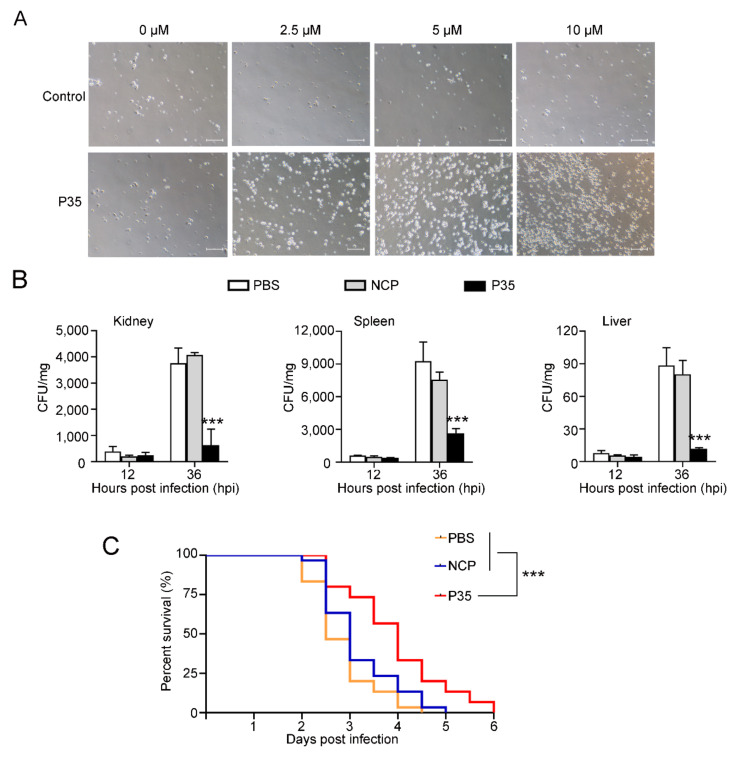
The effect of P35 on PBL migration and flounder immune defense. (**A**) The migration of flounder PBL in the presence of different doses of P35 or the negative control peptide was determined with a transwell migration assay. The migrated cells were observed with a microscope. Scale bar, 20 μm. (**B**) Flounder were infected with *Vibrio harveyi* in the presence of P35, the negative control peptide (NCP), or PBS for different numbers of hours, and bacterial loads (shown as colony forming unit, CFU) in the kidney, liver, and spleen were determined. Values are shown as means ± SD. *** *p* < 0.001. (**C**) The flounder were infected with *V. harveyi* as above and monitored daily for mortality. The significance between the survival of P35-treated fish and control fish was determined with log rank test. n = 30. *** *p* < 0.001.

**Table 1 ijms-23-12722-t001:** The minimal bacterial inhibitory concentration (MIC) of P35 and P17.

Strains	P35 MIC (μM)	P17 MIC (μM)	Strains	P35 MIC (μM)	P17 MIC (μM)
Gram-positive			Gram-negative		
*Bacillus subtilis*	5	—	*Edwardsiella tarda*	40	—
*Bacillus cereus*	—	—	*Escherichia coli*	10	40
*Micrococcus luteus*	10	40	*Pseudomona fluorescens*	20	—
*Staphylococcus aureus*	10	40	*Vibrio anguillarum*	5	20
*Streptococcus iniae*	10	40	*Vibrio harveyi*	10	40

“—“: no inhibition detected at 40 μM.

## Data Availability

All data in the paper are present in the paper or the Appendix A.

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
