# Peer review of "A Non-Canonical Teleost NK-Lysin: Antimicrobial Activity via Multiple Mechanisms"

_ijms, 2022, doi:10.3390/ijms232112722_

Round 1

Reviewer 1 Report

The authors describes two novel NKL peptides P35 and P17 expressed from NKLnc gene from Japanese flounder (Paralichthys olivaceus). The study is quite detailed and well designed, from describing the identification of the NKL from flounder, upregulation of the NKLnc gene in organs of the animal upon infection, expression of the NKLnc in host bacteria cell, to mechanistic studies of their antibacterial mechanism and their effects on immune cells. The authors have demonstrated the potential role of the NKL as part of a pathogen-induced immune response and the confirmed the role of the alpha 2/3 helices of human granulysin to impart bactericidal activity. The novelty of the manuscript is sound and would be of interest to readers in the field.

Author Response

We are thankful to the reviewer for the comments.

Reviewer 2 Report

The paper by Hang Xu and coauthors describes a study of a non-canonical NK-lysin. The analysis of anti-bacterial activity looks comprehensive. However, following critical comments should be addressed before publication.

It is unclear from Introduction if NKL is encoded by all teleost of it is a feature of the mentioned species.
According to Fig.1A, 6 other non-canonical NKL were found earlier. Is there any information about these proteins, their activity and the role? What is known about other species which contain two genes (such as L.crocea or zebrafish)? Some more comments in addition to the mentionned in Discussion would be appreciated.
Did Authors performed a search for potential NKLnc genes in other teleost genomes?
Authors speculate about the role of alpha-helices but do not provide any evidences that the helices are actually formed in the synthesized peptides P35 and P17.

Some details about the methods should be added: how the SS-bonds were predicted and the details about MIC determination.
A separated section of Conclusions would be useful.

Author Response

The paper by Hang Xu and coauthors describes a study of a non-canonical NK-lysin. The analysis of anti-bacterial activity looks comprehensive. However, following critical comments should be addressed before publication.

It is unclear from Introduction if NKL is encoded by all teleost of it is a feature of the mentioned species.

Reply:

The relevant part of Introduction was modified to make the information clear. Lines 61-62.

According to Fig.1A, 6 other non-canonical NKL were found earlier. Is there any information about these proteins, their activity and the role? What is known about other species which contain two genes (such as L.crocea or zebrafish)? Some more comments in addition to the mentionned in Discussion would be appreciated.

Reply:

Of the 6 other non-canonical NKLs in Fig. 6A, only one (Larimichthys crocea NKL) was studied previously, while all other 5 NKLs were identified in this study from NCBI databank, and their functions are unknown. As suggested by the reviewer, more discussion was added with respect to the difference/similarity of different NKL orthologues in zebrafish and Larimichthys crocea. Lines 220-228.

Did Authors performed a search for potential NKLnc genes in other teleost genomes?

Reply:

Yes, we did. As said above, of the 6 other non-canonical NKLs in Fig. 6A, five were identified by us in this study by searching the NCBI databank of teleost genomes.

Authors speculate about the role of alpha-helices but do not provide any evidences that the helices are actually formed in the synthesized peptides P35 and P17.

Reply:

Protein structure prediction indicated that P35 and P17 adopted two and one a helix, respectively. Line 142. Based on the reviewer’s comment, we modified some of the phrases. Lines 253-262.

Some details about the methods should be added: how the SS-bonds were predicted and the details about MIC determination.

Reply:

According to the reviewer’s suggestion, more detailed methods were added to the revised manuscript. Lines 307-310 and Lines 367-372.

A separated section of Conclusions would be useful.

Reply:

According to the reviewer’s suggestion, Conclusion is presented as a separate section. Lines 273-282.